# MAPK Pathway Inhibitors in Thyroid Cancer: Preclinical and Clinical Data

**DOI:** 10.3390/cancers15030710

**Published:** 2023-01-24

**Authors:** Louis Schubert, Mohamed Lamine Mariko, Jérôme Clerc, Olivier Huillard, Lionel Groussin

**Affiliations:** 1Department of Endocrinology, Hôpital Cochin, Assistance Publique-Hôpitaux de Paris, 75014 Paris, France; 2Institut Cochin, Inserm U1016, CNRS UMR8104, Université Paris Cité, 75014 Paris, France; 3Department of Nuclear Medicine, Hôpital Cochin, Assistance Publique-Hôpitaux de Paris, Université Paris Cité, 75014 Paris, France; 4Institut du Cancer Paris CARPEM, Department of Medical Oncology, Hôpital Cochin, Assistance Publique-Hôpitaux de Paris, 75014 Paris, France

**Keywords:** BRAF, MAPK pathway, thyroid cancer, targeted therapy, redifferentiation, radioactive iodine

## Abstract

**Simple Summary:**

The Ras-Raf-MEK-ERK signaling pathway is responsible for regulating cell proliferation, differentiation and survival. Overexpression and overactivation of members within the signaling cascade have been observed in many solid cancers and especially in thyroid cancers. These members are therefore the target of inhibitory therapies, for example tyrosine kinase inhibitors or monoclonal antibodies. These drugs are already used in clinical practice, but their efficacy is not always satisfactory, and they could be subject to escape phenomenon. This is the reason why research is focusing on developing new molecules. We aimed to provide an overview of the MAPK pathway’s physiologic regulation. Furthermore, we summarized the preclinical and clinical studies including redifferentiation studies that used MAPK pathway inhibitors in thyroid cancers.

**Abstract:**

Thyroid cancer is the most common endocrine cancer, with a good prognosis in most cases. However, some cancers of follicular origin are metastatic or recurrent and eventually become radioiodine refractory thyroid cancers (RAIR-TC). These more aggressive cancers are a clinical concern for which the therapeutic arsenal remains limited. Molecular biology of these tumors has highlighted a hyper-activation of the Mitogen-Activated Protein Kinases (MAPK) pathway (RAS-RAF-MEK-ERK), mostly secondary to the *BRAF*^V600E^ hotspot mutation occurring in about 60% of papillary cancers and 45% of anaplastic cancers. Therapies targeting the different protagonists of this signaling pathway have been tested in preclinical and clinical models: first and second generation RAF inhibitors and MEK inhibitors. In clinical practice, dual therapies with a BRAF inhibitor and a MEK inhibitor are being recommended in anaplastic cancers with the *BRAF*^V600E^ mutation. Concerning RAIR-TC, these inhibitors can be used as anti-proliferative drugs, but their efficacy is inconsistent due to primary or secondary resistance. A specific therapeutic approach in thyroid cancers consists of performing a short-term treatment with these MAPK pathway inhibitors to evaluate their capacity to redifferentiate a refractory tumor, with the aim of retreating the patients by radioactive iodine therapy in case of re-expression of the sodium–iodide symporter (NIS). In this work, we report data from recent preclinical and clinical studies on the efficacy of MAPK pathway inhibitors and their resistance mechanisms. We will also report the different preclinical and clinical studies that have investigated the redifferentiation with these therapies.

## 1. Introduction

Thyroid cancer (TC) is the most frequent endocrine cancer with 586,202 new cases worldwide in 2020 [1], with incidence rates that have been rising over the last decades [2]. This can be explained on the one hand by a well-documented overdiagnosis due to increased availability and efficiency of TC screening methods [3] but also by a true increase, especially in the occurrence of advanced-stage TC [4].

Between 90 and 95% of TC derives from thyroid follicular epithelial cells, whereas the remaining develops from C cells resulting in medullary thyroid cancers. Therefore, TCs of follicular origin can be histologically classified into four main groups according to the recent 2022 WHO classification of thyroid neoplasms [5]: differentiated thyroid carcinoma (DTC) including principally papillary thyroid carcinoma (PTC) and follicular thyroid carcinoma (FTC); oncocytic carcinoma; high-grade carcinomas including poorly differentiated thyroid carcinoma (PDTC); and anaplastic thyroid carcinoma (ATC). PTC is the most common type of TC, accounting for 65–93% of cases, and FTC is the second with 6–10% [6]. Both are generally radioiodine avid and have good prognosis when they are treated with total thyroidectomy, selective radioactive iodine (RAI) therapy and thyroid stimulating hormone (TSH)-suppressive therapy [7]. In contrast, ATC is an undifferentiated form of TC with an incidence lower than 1% but the highest mortality rate of all TC due to its radioiodine insensitivity, local aggressiveness and rapid evolution [8].

Constitutive activation of Mitogen-Activated Protein Kinases (MAPK) signaling pathway is frequently observed in the various histological subtypes. Indeed, the MAPK pathway is known to play a major role in the development of many cancers such as melanoma and colon cancer. The most representative proteins of this pathway and also the most important protagonists are RAS, RAF, MEK and ERK. They are involved in various cellular programs such as differentiation, proliferation and apoptosis [9]. Recent genomic studies of thyroid tumors have identified mutually exclusive activating mutations in proteins of this pathway. The main genetic alteration is the activating mutation of *BRAF*, of which the most frequently found genetic event is the *BRAF*^V600E^ hotspot mutation. This mutation is present in 60% of PTCs [10] and 45% of ATCs [11] and is associated with a higher aggressiveness of these cancers. Other mutations could be found to a lesser degree, such as point mutations in the GTPase domain of the genes coding for RAS-isoforms: *HRAS*, *NRAS* and *KRAS* (approximately 13% of PTCs) [10].

Despite good prognosis of DTC, distant metastases (DM) occur in 4–23% of cases, most often in the lung, and one-third of DM patients have no RAI uptake on therapeutic ^131^I whole-body scan (^131^I-WBS), limiting therapeutic possibilities and contributing to poor prognosis and to the majority of deaths associated with DTC [12,13]. These radioiodine refractory (RAIR) cancers represent approximately 5% of all TC [14]. The initial management still consists of thyroidectomy followed by RAI-therapy, but when the diagnosis of RAIR disease is made, they can be treated with multikinases inhibitor with predominant anti-angiogenic activity [15,16]. In 2014 and 2015, sorafenib and lenvatinib showed only partial efficacy, with a progression-free survival (PFS) improvement from 5.8 to 10.8 months and from 3.6 to 18.3 months, respectively, in RAIR-patients included in the phase III DECISION [17] and SELECT [18] trials. Recently, cabozantinib has been tested versus placebo in a phase III trial (COSMIC-311) including progressive RAIR-DTC previously treated with lenvatinib and/or sorafenib. PFS was 11 months with cabozantinib and 1.9 months with placebo [19]. Similarly, first-line treatment of ATC consisting of radiotherapy and chemotherapy remains discouraging with a 1-year overall survival of 20% [20]. These findings led to the search for new therapeutic weapons. Drawing a parallel with melanoma where tyrosine kinase inhibitors (TKIs) targeting MAPK pathway proteins have shown significant efficacy [21], these have been tried in TC. Recently, dabrafenib and trametinib have been incorporated as first-line treatment into the guidelines for management of patients with *BRAF*-mutated ATC [8]. Here, we reviewed the clinical studies on the use of MAPKi in DTC and ATC. We focused on studies and models comprising mutations of *BRAF* and *RAS* as oncogenic drivers as they are the most frequent. *NTRK* and *RET/PTC* gene fusions will not be addressed in this review even though therapeutic advances are very promising in TC due to these molecular abnormalities.

In contrast to melanoma, TC treated with MAPKi is prone to early recurrences secondary to escape phenomena. Several hypotheses have been put forward regarding these resistance mechanisms, which will be described in this review.

New therapeutic perspectives are therefore being explored to overcome this lack of efficacy. On the one hand, a first approach consists of developing more selective and powerful MAPKi in order to achieve a perennial inhibition of the MAPK pathway without the rebound effect. Preclinical and clinical studies on these new therapeutic molecules will be detailed here.

On the other hand, a second approach tries to restore radioiodine avidity of TC by redifferentiating them with MAPKi. In some cases, this enables distant metastasis to concentrate enough radioiodine to administer a new therapeutic dose of iodine-131.

## 2. Physiology of the MAPK Pathway

The MAPK pathway is a privileged pathway in oncogenesis and ensures several vital cellular functions, such as differentiation, proliferation, autophagy and apoptosis [9]. This pathway is schematized in Figure 1 with drugs targeting proteins of this pathway.

The MAPK pathway is composed of serine/threonine and tyrosine kinases, of which the most notable are RAS, RAF, MEK and ERK, and is activated by the stimulation of membrane tyrosine kinase receptors. Specific receptors respond to ligands such as growth factors, for example Fibroblast Growth Factor (FGF) or Epidermal Growth Factor (EGF). After binding of the ligand to the receptor, the latter dimerizes and autophosphorylates, which results in signal transducing into the cell. Its intracellular signaling can be summarized as follows [22]:Recruitment of Growth factor Receptor-Bound protein 2 (GRB2) to the phosphorylated site of the receptor and then attachment of Son Of Sevenless (SOS) to GRB2.SOS, which is a GTP exchange factor, enables the activation of Ras-GDP to Ras-GTP.Ras is a GTPase including three isoforms coded by three genes (*HRAS*, *NRAS* or *KRAS*). It is anchored to the membrane and leads when in active form (Ras-GTP) to the fixation, dimerization and phosphorylation of RAF. The phosphorylation of RAF is not directly performed by RAS but by the SRC Kinase family (SKF) and Casein Kinase 2 (CK2) at the plasma membrane. RAS provides on the one hand the anchoring of RAF to the plasma membrane making it accessible to phosphorylation, and on the other hand, it allows CK2 activation.RAF is a protein kinase of which there are also three isoforms coded by three genes (*ARAF*, *BRAF* and *CRAF*). It activates MEK by phosphorylation on serines 218 and 222.Finally, MEK activates by phosphorylation ERK1 and ERK 2, the two isoforms of ERK. ERK1 is phosphorylated on threonine 202 and tyrosine 204, while ERK2 is phosphorylated on threonine 185 and tyrosine 187.

Activating mutations in *BRAF*, of which the most frequent is the *BRAF*^V600E^ hotspot mutation, render the kinase constitutively active and enable it to signal without dimerization and without being activated by RAS. This mutation is due to the transversion of a thymine (T) to adenine (A) at position 1799 in exon 15, which leads to the replacement of the amino acid valine to glutamic acid, making the kinase domain of the protein functional by modifying its three-dimensional structure [23].

First-generation BRAF inhibitors were therefore developed with the aim of inhibiting mutated BRAF proteins in a targeted manner. Another possibility to inhibit the pathway is to target proteins downstream of BRAF signaling, such as MEK or ERK, which is why MEK inhibitors have emerged.

## 3. RAF and MEK Inhibitors in Clinical Studies of Thyroid Cancer without Redifferentiation Purpose

Several drugs developed by pharmaceutical companies are commonly used in various indications, mainly in oncology. We will not detail the preclinical studies concerning the first generation inhibitors. Three MEK inhibitors (MEKi), binimetinib, cobimetinib and trametinib, as well as three first-generation RAF inhibitors (RAFi), dabrafenib, vemurafenib and encorafenib, have been FDA-approved in *BRAF* mutated metastatic melanoma. The MEKi selumetinib has no approval in cancers but is indicated for type 1 neurofibromatosis with symptomatic, inoperable plexiform neurofibromas. For TC, only dabrafenib and trametinib have been FDA-approved and are indicated for locally advanced or metastatic ATC with *BRAF*^V600E^ mutation and with no satisfactory locoregional treatment options. This approval followed Subbiah’s phase II trial [24] where 36 patients with locally advanced or metastatic *BRAF*^V600E^ ATC were treated with dabrafenib and trametinib. Objective response rate (ORR) was 56% with three complete responses (CR). Progression-free survival (PFS) and overall survival (OS) were, respectively, 6.7 and 14.5 months improving the prognosis of these tumors without therapeutic alternative to conventional chemoradiotherapy. The most frequent treatment related adverse events with these two molecules were pyrexia (47%), anemia (36%), decreased appetite (33%), fatigue (33%) and nausea (33%), while any grade 3/4 adverse events occurred in 58% of patients. This dual therapy is now recommended by the European Society for Medical Oncology (ESMO) [25] and the American Thyroid Association (ATA) [8] as first-line therapy for these patients, and those with a significant response in the neck may be considered for surgery to remove the primary tumor and/or locoregional disease.

In contrast, MAPKi have not yet been approved by the European and American authorities for DTC and are therefore used off-label. Their anti-tumor activity has been studied in five phase I-II clinical trials (Table 1).

Four studies were interested in BRAF inhibitors as monotherapy. Vemurafenib was tested in two studies, a phase I and a phase II study in metastatic or recurrent *BRAF*-mutated PTCs and showed an ORR ranging from 27.3 to 38.5% [26,27]. The phase II study had the largest number of patients (n = 51), and two cohorts were formed, one naive and one already treated with multi-kinase therapy targeting VEGFR. The PFS and OS in the previously treated cohort were 8.9 and 14.4 months, respectively, whereas in the naïve cohort, the PFS was 18.2 months. Serious adverse events were reported for 62% of patients in the naive cohort and 68% of patients in the pretreated cohort. The more frequent serious adverse events reported were cutaneous squamous cell carcinoma (27% in the naive cohort, 20% in the pretreated cohort), keratoacanthoma (8% in the naïve cohort 1, 12% in the pretreated cohort), followed by dyspnea, pneumonia, hypotension, cerebrovascular accident and squamous cell carcinoma [27]. Dabrafenib was tested in a phase I study in metastatic *BRAF*-mutated PTC/PDTC and showed an ORR of 29% with only partial responses (PR) [28]. It was also studied in combination with trametinib in a randomized phase II trial conducted by Busaidy et al. with *BRAF*-mutated RAIR PTC. A total of 26 patients were included in the dabrafenib alone group with ORR of 35%, and 27 patients were included in the dabrafenib and trametinib group with ORR of 30%. In the arm treated with monotherapy, PFS was 10.7 months and OS was 37.9 months, whereas in the arm treated with bitherapy, PFS was 15.1 months and OS was 47.5 months [29]. Concerning adverse events, any grade treatment-related adverse events were noted in 100% of patients in each arm and were predominantly grade 1 or 2. Grade 3 treatment-related adverse events were noted in 15 patients (58%) on dabrafenib versus 13 patients (48%) on dabrafenib and trametinib. There were no grade 4 or 5 treatment-related adverse events, but the number of treatment-related serious adverse events were greater with dabrafenib and trametinib than with dabrafenib alone (78% versus 35%). Finally, only one study looked at a MEK inhibitor in monotherapy, namely selumetinib, but the 32 RAIR PTC patients included were not all *BRAF*-mutated (20 *BRAF*-Wild-type patients). The ORR was 3%, and the reported PFS was 8 months [30].

Even if melanomas and thyroid cancers both have a hyperactivation of the MAPK pathway, they remain two different diseases, and comparing the efficacy of MAPKi between these two cancers is not easy. However, some points are noteworthy and deserve to be highlighted. BRAFi used as monotherapy seems to have a higher response rate in melanoma with an ORR varying in five phase III trials between 40 and 51% [31,32,33,34,35] in contrast to the previously discussed studies in DTC with an ORR between 27.3 and 38.5%. In addition, the four phase III trials in melanoma comparing BRAFi and MEKi to BRAFi alone showed significantly better efficacy of dual therapy than monotherapy at least on one of these three robust end points that are ORR, PFS and OS [31,32,34,35]. In contrast, Busaidy’s trial which looked at the same question in RAIR-PTC showed a trend in efficacy for the dual therapy, but it was not significant. However, it was a phase II study, and the small number of patients (53 patients) may explain a lack of power.

These poorer results in thyroid cancers have led researchers to wonder about the existence of primary resistance or escape mechanisms to these MAPKi.

## 4. Mechanisms of Resistance to MAPKi

Resistance mechanisms can be classified into two broad categories, primary or secondary, depending on whether they are already present or whether they are acquired after treatment with TKIs. Primary or intrinsic resistance is defined by lack of clinical benefit upon initiation of treatment, whereas secondary or acquired resistance is defined by the occurrence of progressive disease after an initial period of clinical response.

Moreover, the molecular mechanisms underlying the resistance phenomena also make it possible to classify them, and we have chosen to present them in this way in Table 2.

A frequent type of resistance mechanisms is genomic instability. The constant modification of the genome under the pressure of TKIs allows the emergence of point mutation or copy number variation on genes regulating the survival or proliferation of tumor cells. Danysh et al. showed in a *BRAF*^V600E^ PTC cell line that an acquired mutation of *KRAS*^G12D^ leads to vemurafenib resistance via activation of the MAPK and PI3K/AKT pathways [38]. Activating mutations on different RAS isoforms were also confirmed on blood biopsies or on progressive lesions in two PTC and two ATC patients who progressed after BRAFi treatment [40]. Moreover, PIK3CA has also been shown to be an oncogenic driver in Landa’s work describing ATC molecular abnormalities in humans, as 15% of ATC tumors analyzed had both PIK3CA and BRAF mutations [11]. Transgenic mice developed with a *PIK3CA*^H1047R^ and *BRAF*^V600E^ double mutation by Roelli *et al.* showed primary resistance to PLX-4720 (BRAFi) compared to *BRAF*^V600E^ mutant mice that were sensitive [36]. Genetic alterations in genes regulating apoptosis and cell cycle were also found. Duquette et al. [37] and Antonello et al. [39] described, respectively, on vemurafenib resistant cell models, a copy number loss of Cyclin Dependent Kinase Inhibitor 2A (*CDKN2A*) and *de novo* mutations in the RNA-binding motifs (*RBM*) genes family. These two gene families are known to be important players in regulation of the cell cycle in response to DNA-damage. In addition, Antonello finds that dual therapy with BRAFi and palbociclib, which is a CDK4/6 inhibitor commonly used in breast cancer, overcomes the resistance mechanism and is more effective than vemurafenib alone. Duquette *et al.* also suggested that a copy number gain of myeloid cell leukemia-1 (*MCL1*), which is anti-apoptotic, would result in an impairment of the BCL2-regulated apoptotic pathway allowing tumor cells to survive. Finally, Bagheri et al. demonstrated that Rac Family Small GTPase 1 (*RAC1*) mutation and copy number gain lead to acquired resistance via changes in cell adhesion properties and cell proliferation [41]. Indeed, the *RAC1/PAK1* pathway is implicated in many cellular processes, including cell cycle, cell–cell adhesion, motility through cytoskeletal reorganization and cell growth through activation of the signaling pathway.

Autocrine secretions by tumoral cells have been suggested to be responsible for acquired resistance to BRAFi. Autocrine loops may reactivate the MAPK pathway or recruit another pathway involved in cell proliferation. A study with a xenograft mouse model and a study with transgenic *p53* deletion and *BRAF*^V600E^ mouse models showed that the initial inhibition of the MAPK pathway led to the secretion of Hepatocyte Growth Factor (HGF) which consequently stimulates the overexpressed c-Met receptor [42,43]. The result was a reactivation of the MAPK and PI3K/AKT pathways. These same pathways were also reactivated in Montero-Conde’s study, secondary to neuroregulin-1 (NRG1) secretion and stimulation of the overexpressed HER3 membrane receptor [44]. This mechanism of resistance was overcome by the combined use of vemurafenib and lapatinib which is a HER2 inhibitor used in metastatic breast cancer overexpressing HER2. The last two studies about autocrine loops looked at STAT3/JAK pathway activation following the inhibition of *BRAF*^V600E^ PTC and ATC cell lines by vemurafenib. This pathway also has a privileged role in controlling various cellular functions favorable to tumorigenesis and is interconnected with the MAPK and PI3K/AKT pathways. Sos *et al.* [45] and Notarangelo *et al.* [46] demonstrated that autocrine secretion of IL-6, which is a JAK receptor tyrosine kinase ligand, led to the activation of this pathway and finally to resistance. Tofacitinib and tocilizumab, which are, respectively, JAK and IL6-R inhibitors commonly used for various indications, can counteract this escape phenomenon in dual therapy with BRAFi.

Other authors have shown that overexpression and/or overactivation of proteins that are not main actors but rather modulators of the MAPK or PI3K/AKT pathways, can decrease the efficacy of MAPK inhibitors. Tribbles homolog 2 (TRIB2) [47], Epidermal growth factor receptor (EGFR) [48] and Src homology 2 containing protein tyrosine phosphatase 2 (SHP2) [49] are some examples. TRIB 2, a member of the tribbles family, is a scaffold protein that can interact with E3-ubiquitin ligases and control protein stability of downstream effectors. SHP2 is a protein phosphatase that facilitates signal transduction from membrane receptors to early effectors of cell signaling pathways.

Less typical mechanisms of resistance in oncology have been proposed. The first one is related to cancer stem cells (CSCs). This is a relatively small cell sub-population within tumor mass with stem-cell-like properties and the ability to grow as non-adherent spheroids and to sustain self-renewal. Chemotherapy resistance has already been reported due to CSCs [53]. In his study, Giani et al. founds that vemurafenib resistance of this cell sub-population may be linked to tumor-progression-locus-2 (TPL2) protein upregulation [50]. This protein is also known as MAP3K8, which is a mitogen-activated protein kinase activated downstream of TNFαR, ILR, TLR and GPCRs and which regulates the MEK1/2 and ERK1/2 pathway. The second resistance mechanism reported by Hu et al. was mediated by oxidative stress and the Redox factor-1 (Ref-1) protein [51]. It is also known as apurinic/apyrimidinic endonuclease 1 (APE1) and is a highly conserved functional enzyme that has a redox function that regulates the activity of a variety of important transcription factors. It also has nucleic acid endonuclease activity, allowing Ref-1 to function as a DNA repair enzyme leading to pro-survival signals. The last mechanism, reported by Run et al. would involve an increase in High Mobility Group Box 1 (HMGB1) mediated autophagy [52]. This is a highly conserved and ubiquitous non-histone chromosomal protein that organizes DNA and regulates transcription.

A final mechanism of resistance that has been mostly proven in *BRAF*-mutated models of melanoma, lung cancer and colon cancer deserves to be highlighted. It goes through the signaling of RAF isoforms and allows a paradoxical reactivation of the MAPK pathway. Indeed, it has been proven that most clinically available RAFi, including dabrafenib and vemurafenib, inhibit the MAPK pathway by binding its catalytic site in an ATP competitive manner and blocking the BRAF^V600E^ monomer in a certain allosteric conformation. Despite this initial inhibitory capacity, these RAFi induce dimerization of drug bound BRAF^V600E^ with CRAF or ARAF, leading to downstream signalization through primed CRAF or ARAF monomers that cannot bind RAFi for allosteric reasons [54,55,56,57]. This signaling through the dimeric form of RAF is probably not a resistance mechanism in its own right, but rather corresponds to the mechanism by which the pathway can refunction in response to the several stimuli mentioned above (activating genetic alteration, autocrine secretion...) despite the inhibition by BRAFi. This is called paradoxical activation of the MAPK pathway [23]. Based on these findings, new RAFi have been developed and RAFi are now classified in two groups, the first generation RAFi, also called type 1 RAFi which can only inhibit the BRAF mutated monomer, and the second generation RAFi or Pan-RAF inhibitors capable for chemical and allosteric reasons not detailed here, to inhibit the signaling of the BRAF mutated monomer but also the signaling of dimers. Pan-RAF inhibitors are still in the preclinical stage, and we will detail hereafter two studies conducted on thyroid models.

## 5. New Treatments Perspectives

Given the multiple mechanisms of resistance through a paradoxical activation of the MAPK pathway, we have listed the different studies on new molecules targeting this pathway in preclinical models of TC. They are listed in Table 3 according to their target.

We will try to illustrate these preclinical data with early clinical trials testing these new molecules, when there are some available.

As discussed above, second generation BRAFi were developed to inhibit RAF signaling in both monomeric and dimeric forms. Two molecules, TAK-632 and LY3009120, were tested and compared to vemurafenib. Both molecules first proved to be effective in inhibiting RAF dimers by avoiding reactivation of the pathway. Then, they showed superiority over vemurafenib in all experiments performed. The experiments included cell and mouse models for LY3009120 [58] and only cell models for TAK-632 [56]. A phase I study looked at LY3009120, including mostly *RAS* or *RAF*-mutated non-squamous cell lung carcinoma (NSCLC) or colorectal cancer (CRC) but no TC. Even if a phase I trial is not designed for efficacy assessment, no CRs or PRs were observed in the study. Stable disease (SD) was observed in 8 patients out of 34. Of the 8 patients who had SD, 5 patients had *BRAF* mutations (among a total of 12 BRAF cancers), 2 had *KRAS* mutations (among a total of 17 KRAS cancers), and 1 had *NRAS* mutation (among a total of 5 *NRAS* cancers) [60]. Another phase I trial was conducted with a pan-RAFi, lifirafenib (BGB-283), and was more encouraging. The disease control rates (CR and PR and SD) on 53 *BRAF*-mutated and 66 *RAS*-mutated patients were, respectively, 67.9% and 53%. In the *BRAF*-mutated cohort, one patient with melanoma achieved CR and eight patients had PR (five melanoma, two PTC, one low-grade serous ovarian cancer). Only two patients in the *RAS*-mutated cohort had PR (one endometrial cancer and one NSCLC). There were 33 SD, and no response was seen in patients with colorectal cancer (n = 20). It is important to note that five PTC had been included; two had PR, and three had SD [61].

Finally, the most downstream protein in the MAPK pathway was also targeted. An ERK1/2 inhibitor, SCH772984, was tested in dual therapy with dabrafenib. In five *BRAF*^V600E^ cell lines (ATC and DTC), dual therapy showed superiority over dabrafenib alone on cell assays and also on tumor growth inhibition in vivo [59]. SCH772984 has not been tested in humans but another ERKi, Ulixertinib (BVD-523), was explored by Sullivan et al. in a phase I trial including mostly melanoma, NSCLC and CRC. PRs were seen in 11 of 81 (14%) evaluable patients, including 3 of 18 with *NRAS*-mutant melanoma, 3 of 12 with *BRAF*-mutant lung, 1 of 15 with BRAF/MEK inhibitor–refractory *BRAF*^V600E^ mutant melanoma, and 4 of 21 with other BRAF-mutant cancers. Six TC were enrolled, but specific data were not available [62].

It is noteworthy that no RASi has been tested in preclinical or clinical studies regarding TC. Yet, sotorasib, a specific and selective inhibitor of *KRAS^G12C^* recently approved by the FDA in 2021, demonstrated clear progress in management of locally advanced/metastatic NSCLC. Indeed, in a study that included 124 patients with *KRAS^G12C^*-mutant NSCLC who had previously received other treatments, sotorasib showed an ORR of 37.1% with median duration of response of 11.1 months [63]. In contrast, standard therapies shrink tumors in less than 20% of people with NSCLC that has come back after previous treatment, and those effects are usually short-lived [64].

## 6. Iodine Recaptation Approach in Thyroid Cancers Models

In physiology, iodide is concentrated from the blood stream into the thyroid follicles through the action of the NIS and then incorporated into thyroglobulin, a process referred to as organification. It is facilitated by various enzymes, the most important of which is thyroperoxidase (TPO), and modulated by TSH level. These two crucial steps, uptake and organification, can be impaired in RAIR-TC. The comprehensive characterization of 496 papillary thyroid cancers published in 2014 by the Cancer Genome Atlas (TCGA) Research Network highlighted two main categories of TC. The first was “BRAF-like TC”, composed of mostly PTC and characterized by a low degree of differentiation with downregulation of genes necessary for proper iodine metabolism such as *NLC5A5* (coding for NIS), *TPO* and *TG*. The second was “RAS-like TC”, composed of mostly follicular cancers, keeping a better differentiation and expression of *NLC5A5*, *TG* and *TPO* [10]. Other studies suggested that the NIS protein is present in the intra-cellular compartments in some thyroid cancer tissues but is not transported to the cell membrane, explaining why it is not biologically active [65].

The NIS is therefore the cornerstone actor of an effective RAI-therapy. The various preclinical studies that we reviewed have therefore investigated NIS expression, both at the transcriptional level with quantification of mRNA assessed by RT-qPCR and at the protein level by NIS Western blotting (WB). NIS localization at the membrane was also studied using fluorescence microscopy. Finally, its functionality, reflecting the achievement of all the previous steps, could have been evaluated by quantifying the incorporation of iodine in cell or mouse models by radioactivity assay. The studies are summarized in Table 4.

Three studies looked at the efficacy of BRAFi or MEKi as monotherapy, one in mouse models [68] and two in cell lines [66,67]. Each study found an NIS re-expression at the mRNA and protein level. The increase of iodine uptake capacity was found by Bonaldi et al. in a cell model [66] and by Nagarajah et al. in a mouse model [68]. In the latter study, a tumor response after RAI-therapy in mice was observed after treatment with MEKi.

One study investigated the re-expression of NIS after treatment with the combination of dabrafenib and trametinib in PTC-derived primary cell cultures and it was more successful than dabrafenib or trametinib alone [69].

Three studies investigated in *BRAF*-mutated PTC cell lines, a dual therapy with BRAFi or MEKi combined with a new targeted therapy. All the studies investigated NIS expression, NIS cell membrane localization and iodine uptake capacity. The first investigated panobinostat, a histone deacetylation (HDAC) inhibitor, because iodine-metabolizing gene silencing has been related to histone deacetylation [70]. In a similar way concerning epigenetic modifications, the second study investigated tazemetostat which is a histone methyltransferase inhibitor. Certain types of histone methylation (e.g., histone H3 lysine 27 trimethylation modification) lead to depression of gene expression through enhancer of zeste homolog 2 (EZH2), a critical methyltransferase and an epigenetic mark for the maintenance of gene silencing [71]. The third study looked at lapatinib, an HER inhibitor that we described earlier [72]. All three studies found a higher NIS expression and membrane localization as well as a greater functionality with the bi-therapy comporting the new molecule and one of the MAPKi compared to MEKi or BRAFi alone [70,71,72]. A last study focused on the TGF-β/SMAD signaling pathway [73] because *BRA*F^V600E^-induced suppression of NIS expression was shown to be partly mediated by transforming growth factor 1 (TGFB1) in an MEK-independent manner [74]. Luckett *et al.* tested vactosertib, an inhibitor of transforming growth factor receptor (TGFBR) and activin A receptor type 1B (ACVR1B), in a *BRAF*^V600E^-mutated mouse model. ACVRB1 is also a membrane receptor stimulable by a protein of the TGF beta superfamily, the activin, leading to the activation of SMAD proteins. Bi-therapy involving CKI, a MEKi and vactosertib showed enhanced iodine uptake in thyroid tumors compared to CKI alone [73].

These encouraging preclinical data have allowed MAPKi to be tested in human clinical trials.

## 7. Clinical Redifferentiation Strategies in Radioactive Iodine Refractory Thyroid Cancers

Thanks to their redifferentiating property, MAPKi have been studied in clinical trials for the restoration of iodine avidity in RAIR-TC. The aim is to have RAI-therapy back as a therapeutic weapon, knowing that systemic therapies with TKIs such as anti-angiogenic have inconsistent effectiveness and frequent side effects. Eight publications on this topic were analyzed and are presented in Table 5.

The majority of patients included in the studies were *BRAF*-mutated PTCs but there was a significant proportion of Wild Type (WT) or *RAS*-mutated cancers and PDTCs histology [77,79,81]. It is therefore difficult to make a systematic comparison between these studies. For the objective of homogenization, only studies with multiple patients were presented and case reports were not reviewed. Nevertheless, the case report by Leboulleux et al. deserves to be cited as it is the proof of redifferentiating efficacy in humans of a bi-therapy with dabrafenib and trametinib. Indeed, a 59-year-old patient treated with this dual therapy for *BRAF*^K601^-mutated metastatic PDTC developed thyrotoxicosis and became RAI avid again. Moreover, a biopsy performed two months after treatment initiation showed a more differentiated growth pattern with a microfollicular appearance and intraluminal colloid material in addition to an NIS re-expression compared to baseline [83].

The redifferentiating therapies used in the reviewed studies were BRAFi, such as dabrafenib or vemurafenib, MEKi, such as selumetinib or trametinib, or a BRAFi and MEKi combination, essentially dabrafenib and trametinib. The duration of redifferentiating therapy before RAI avidity assessment was generally short, ranging from 3 to 6 weeks. Only the retrospective study by Jaber et al. included patients treated with long-term MAPKi up to 76.4 months [77]. In fact, two strategies concerning the duration of redifferentiation treatment exist: short treatment duration of a few weeks to allow redifferentiation and to limit the side effects of these therapies and long-term treatment strategy to benefit from an additional cytotoxic effect. These two strategies have to be discussed according to patient profiles. Short course treatment could be adapted to comorbid patients or to those with a low tumor burden, whereas long-term treatment seems to be appropriate for patients with a high tumor burden, assuming that TKIs could have additional antitumor/cytotoxic effects.

The reviewed studies were heterogeneous in the modalities of redifferentiation but had some commonalities. Their primary end point was the same, that is the rate of patients with RAI uptake restoration evaluated by diagnostic I^123^-WBS, I^131^-WBS or I^124^ PET-scan. Then only patients who were RAI avid again on diagnostic evaluation were re-treated with RAI-therapy. The rate of RAI uptake restoration on diagnostic examination was between 60 and 67%. Two studies are to be analyzed in their own right. The first is Weber’s study, which found a rate of 35%, but this can be explained by the inclusion of 70% of WT-TC [81]. The second is Tchekmedyian’s study which showed a high response rate of 83%. It is the only study to use a dual therapy with a MAPKi and an anti-HER3 monoclonal antibody named CDX-3379 [82].

A different design is noteworthy in the MERAIODE trial, where all included patients were re-treated with RAI-therapy, whether avid or not at diagnostic nuclear assessment. Thus, the rate of RAI uptake restoration could have also been calculated on post-therapeutic WBS and a difference of 32% between post-therapeutic WBS and diagnostic WBS was found (95% vs. 65%), suggesting that this rate varies according to the activity administered [80]. Indeed, the administered activities for diagnostic examinations rarely exceed 370 MBq, whereas the administered activity for treatment can go up to 11 GBq in all the presented studies. This relationship between administered activity and uptake was also illustrated in a case report where an RAIR metastatic TC with *RET-*fusion treated with selpercatinib had more RAI avid locations on the post-therapeutic WBS than on the diagnostic WBS [84].

Finally, due to the heterogeneity of the studies, the RECIST criteria were difficult to compare. The ORR ranged from 14% to 75% and was generally evaluated early, at 6 months or at 1 year maximum. However, the majority of patients presented a stable disease, meaning that the redifferentiation strategy was at least partially effective, but a long-term evaluation is lacking to judge the effectiveness over time of these punctual treatments.

Another interrogation is the impossibility to distinguish between tumor response resulting from a cytotoxic effect of MAPKi and the effect of RAI-therapy after the restoration of radioiodine uptake. Antitumor effect of the TKIs will be at least partially evaluated in MERAIODE for which we are waiting for the complete results. Nevertheless, a short-term treatment is unlikely to produce long-term and persistent effect.

However, in view of the encouraging global effects of these therapeutic strategies, the prospect of giving an adjuvant MAPKi for primary RAI treatment to RAI avid TC in order to boost the dose delivered to the tumors was considered by Ho et al. This phase III placebo-controlled trial, named ASTRA, included 233 high-risk DTC patients (primary tumor > 4 cm; gross extrathyroidal extension outside the thyroid gland [T4 disease]; or N1a/N1b disease with ≥ 1 metastatic lymph node(s) ≥ 1cm or ≥ 5 lymph nodes [any size]) who were randomly assigned 2:1 to selumetinib or placebo for approximately 5 weeks followed by 3.7 GBq RAI-therapy. The effectiveness of this approach was not demonstrated as no statistically significant difference in complete response rate 18 months after RAI was found (40% for selumetinib versus 38% for placebo; OR = 1.07 [95% CI, 0.61 to 1.87]; *p* = 0.8205) [85]. However, this lack of effectiveness could be explained by difficulties in maintaining full dose selumetinib continuously due to drug toxicity and also by the drug tested itself. Indeed, selumetinib alone may not be the best treatment when RAF/RAS status of the patient is unknown because BRAF-targeted drugs are superior to MEK inhibitors for redifferentiation in BRAF-mutant RAIR patients.

## 8. Conclusions

*BRAF^V600E^* is the most common molecular alteration in thyroid cancers and is associated with tumor aggressiveness and poor prognosis because of the continuous activation of the MAPK pathway leading to cell proliferation and survival.

Therapies targeting key players of the MAPK pathway such as RAF and MEK have shown encouraging results in thyroid cancers, but less impressive than in melanoma where the MAPK pathway is also at the basis of oncogenesis. It is therefore necessary to understand differences in efficacy observed between thyroid cancers and other cancers with activation of the MAPK pathway, as well as mechanisms of resistance to MAPKi.

Drug resistance can occur due to genomic instability with the proliferation of pre-existing resistant clones harboring intrinsic mutations or the occurrence of new genetic and epigenetic alterations, which often activate molecules up/downstream from the MAPK pathway. Several other resistance mechanisms have been identified and induce a MAPK pathway paradoxical activation or the recruitment of another proliferation signaling pathway.

There is therefore a growing need to develop and test in preclinical studies new molecules targeting the MAPK pathway, such as pan RAF inhibitors, RAS or ERK inhibitors. There is also a necessity to explore drug combinations as novel strategies to overcome single-agent-induced resistance. These combinations could include multiple MAPKi or one MAPKi with a promising drug targeting another proliferation pathway or counteracting a specific mechanism of secondary resistance.

Clinical studies must test molecules which show promising results in preclinical studies, with a rigorous methodology and try to include a large number of patients.

The redifferentiation strategies are very original and could be useful in thyroid cancers, particularly in refractory metastatic diseases or as adjuvant therapy in diseases at high risk of recurrence. These strategies provide a combined and perhaps synergic antitumor effect of the targeted therapy and the RAI-therapy. One of the major challenges is to first identify cancers accessible to these strategies and then to choose the most effective inhibitor or combination of drugs that will be used. Thyroid cancers not accessible to redifferentiation strategies must be studied as a separate entity and be better characterized at the molecular level.

## Figures and Tables

**Figure 1 cancers-15-00710-f001:**
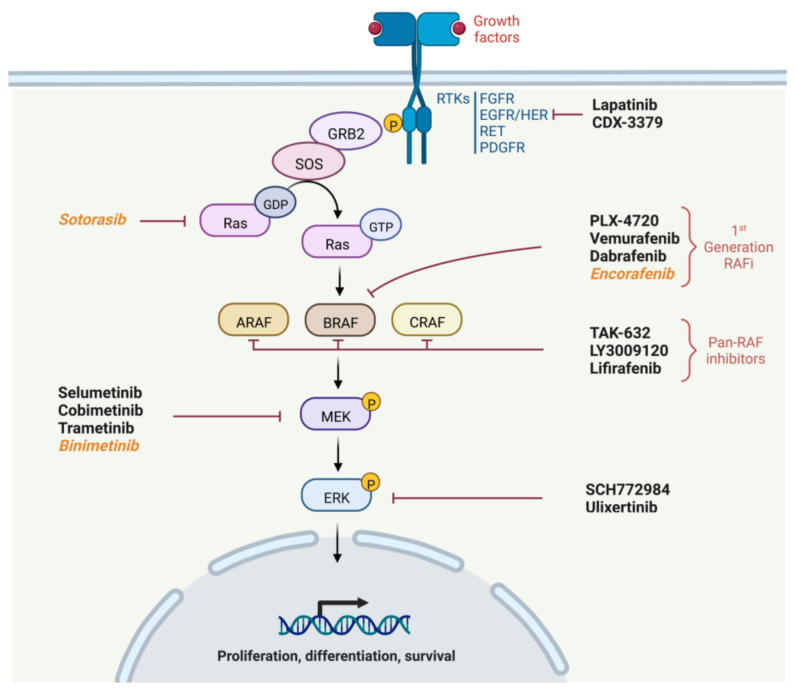
MAPK signaling pathway and TKI targets. Legend: molecules never used or tested in TC are written in italic. Created with BioRender.com.

**Table 1 cancers-15-00710-t001:** Clinical trials of RAF and MEK inhibitors in thyroid cancers.

Thyroid Cancer Types	Drug Targets	Therapies	Patients Number	Study Design	ORR	Median Duration of Response (Months)	Median PFS (Months)	Median OS (Months)	Ref
Locally advanced or metastatic BRAF mutated ATC	BRAF + MEK1/2	Dabrafenib + trametinib	36	Open-label, phase II trial	56% (3 CR, 17 PR)	12-months DoR: 50%	6.7	14.5	[24]
Metastatic BRAF mutated PTC	BRAF	Vemurafenib	3	Phase I trial	33.3% (1 PR)	NA	n1 = 11.4 n2 = 11.7 n3 = 13.2	n1 = 15 n2 = 21 n3 = 31.7	[26]
Metastatic or recurrent BRAF mutated PTC	BRAF	Vemurafenib	Total: 51 Naïve, cohort 1 (C1): 26 Previous TKI, cohort 2 (C2): 25	Open-label, phase II trial	C1: 38.5% (10 PR) C2: 27.3% (6 PR)	C1: 16.5 C2: 7.4	C1: 18.2 C2: 8.9	C1: NR C2: 14.4	[27]
Metastatic BRAF mutated PDTC or DTC	BRAF	Dabrafenib	14	Phase I trial	29% (4 PR)	NA	11.3	NA	[28]
BRAF Mutated RAIR PTC	BRAF	Dabrafenib	26	Randomized phase II trial	35% (9 PR)	18.3	10.7	37.9	[29]
BRAF Mutated RAIR PTC	BRAF + MEK1/2	Dabrafenib + trametinib	27	Randomized phase II trial	30% (8 PR)	17.0	15.1	47.5	
BRAF Mutated or WT RAIR PTC	MEK1/2	Selumetinib	32	Open-label, phase II trial	3% (1 PR)	NA	8	NA	[30]

Abbreviations: CR, complete response; DoR, duration of response; PR, partial response; NA, non-applicable; NR, non-reported; ORR, objective response rate; OS, overall survival; PFS, progression-free survival; TKI, tyrosine kinase inhibitor; WT, wild-type.

**Table 2 cancers-15-00710-t002:** Mechanisms of resistance to MAPKi in thyroid cancer models.

Type of Resistance Mechanism	Drugs Used to Study Resistance (Target)	Thyroid Cancer Models	Mechanism of Resistance (Intrinsic or Acquired Resistance)	Drug Used to Overcome Resistance (target)	Resistance Overcome	Ref
Genomic instability	PLX-4720 (BRAF)	- BRAF^V600E^ ATC cell line - BRAF^V600E^ and double mutant BRAF^V600E^ + PIK3CA^H1047R^ TC mouse models	PIK3CA^H1047R^ mutation (intrinsic resistance) leading to: - MAPK pathway paradoxical activation	GDC-0941 (PIK3CA)	Yes	[36]
	Vemurafenib (BRAF)	- BRAF^V600E^ PTC cell lines - Samples derived from BRAF-mutated PTC patient - Primary cell culture of BRAF^V600E^ metastatic or recurrent PTC	Copy number gain of MCL1 and loss of CDKN2A (intrinsic resistance) leading to: - Impairment of the BCL2-regulated apoptotic pathway - CDK4/6 pathway activation	Obatoclax (BCL2/MCL1)	Yes	[37]
		BRAF^V600E^ PTC cell line	KRAS^G12D^ mutation (acquired resistance) leading to: - PI3K/AKT pathway activation - MAPK pathway paradoxical activation	NA	NA	[38]
		BRAF^V600E^ PTC cell line	Amplification of chromosome 5 and de novo mutations in the RBM genes family (intrinsic and acquired resistance) leading to: - Chromosome instability and deregulation of cell cycle checkpoints in response to DNA-damage	Palbociclib (CDK4/6)	Yes	[39]
	Vemurafenib (BRAF) Dabrafenib (BRAF) + Trametinib (MEK)	2 PTC patients and 2 ATC patients with BRAF mutation	Acquired KRAS^G12V^ (n = 2), NRAS^Q61K^ (n = 1), and NRAS^G13D^ (n = 1) mutations on progressive metastatic lesions after treatment with MAPKi	NA	NA	[40]
	Dabrafenib (BRAF)	- BRAF^V600E^ PTC cell lines - PTC BRAF-mutated patient - Patient derived cell line	RAC1 mutation and copy number gain (acquired resistance) leading to: - RAC1/PAK1 pathway activation	EHop-016 (RAC1)	Yes	[41]
Autocrine loop	PLX-4720 (BRAF)	- Transgenic p53- and BRAF^V600E^ ATC mouse model - Mouse derived cell lines	c-Met overexpression and HGF increased secretion (acquired resistance) leading to: - PI3K/AKT pathway activation - MAPK pathway paradoxical activation	PF-04217903 and crizotinib (c-Met)	Yes	[42]
	Vemurafenib (BRAF)	- BRAF^V600E^ PTC and ATC cell lines - ATC xenograft mouse model	c-Met overexpression and HGF increased secretion (acquired resistance) leading to: - PI3K/AKT pathway activation	PHA665752 (c-Met)	Yes	[43]
		BRAF^V600E^ PTC and ATC cell lines	HER3 overexpression and activation by NRG1 secretion (acquired resistance) leading to: - PI3K/AKT pathway activation - MAPK pathway paradoxical activation	Lapatinib (HER)	Yes	[44]
Autocrine loop	Vemurafenib (BRAF)	BRAF^V600E^ PTC and ATC cell lines	IL6 secretion (acquired resistance) leading to: - STAT3/JAK pathway activation	Tofacitinib (JAK)	Yes	[45]
				Tocilizumab (IL6-R)		[46]
Upregulation of proteins operating synergistically with the MAPK and PI3K/AKT pathways	Vemurafenib (BRAF)	BRAF^V600E^ PTC cell lines	TRIB2 upregulation induced by activation of the Wnt/β-catenin pathway (acquired resistance) leading to: - PI3K/AKT pathway activation - MAPK pathway paradoxical activation	ICG-001 (β-catenin)	Yes	[47]
		BRAF^V600E^ PTC and ATC cell lines	EGFR overactivation (acquired resistance) leading to: - PI3K/AKT pathway activation - MAPK pathway paradoxical activation	Gefitinib (EGFR)	Yes	[48]
	Selumetinib (MEK)	- BRAF^V600E^ PTC cell lines - PTC xenograft mouse models - Transgenic BRAF^V600E^ mouse models	SHP2 upregulation and activation (acquired resistance) induced by upregulation and activation of multiple RTKs (RET, FGFR, HER2…) leading to: - MAPK pathway paradoxical activation	SHP099 (SHP2)	Yes	[49]
Cancer Stem Cells (CSCs) mediated resistance	Vemurafenib (BRAF)	CSCs selected from BRAF^V600E^ ATC cell lines	TPL2 overexpression in CSCs (acquired resistance) leading to: - PI3K/AKT pathway activation - MAPK pathway paradoxical activation	(TPL2)	Yes	[50]
Oxidative stress mediated resistance	Vemurafenib (BRAF)	- BRAF^V600E^ PTC cell lines - Samples derived from BRAF mutated PTC patient	Ref-1 upregulation (intrinsic resistance) leading to: - MAPK pathway paradoxical activation	E3330 (Ref-1)	Yes	[51]
Autophagy mediated resistance	Vemurafenib (BRAF)	BRAF^V600E^ PTC Cell line	HMGB1 upregulation (acquired resistance) leading to: - HMGB1-induced autophagy	3-MA (Autophagy inhibitor)	Yes	[52]

Abbreviations: AKT, ak strain transforming; BCL2, B-Cell CLL/Lymphoma 2; CDK4/6, cyclin dependent kinase 4/6; CDKN2A, cyclin dependent kinase inhibitor 2A; CSCs, cancer stem cells; EGFR, epidermal growth factor receptor; FGFR, fibroblast growth factor receptor; HER3, human epidermal growth factor receptor 3; HGF, hepatocyte growth factor; HMGB1, high mobility group box 1; IL6, interleukin-6; MCL1, myeloid cell leukemia-1; NA, non-applicable; NRG1, neuroregulin-1; PAK1, P21 activated kinase 1; PIK3CA, phosphatidylinositol-4,5-bisphosphate 3-kinase catalytic subunit alpha; RAC1, rac family small GTPase 1; RBM, RNA-binding motifs; Ref-1, redox factor-1; RET, rearranged during transfection; RTKs, receptor tyrosine kinases; SHP2, SH2 containing protein tyrosine phosphatase-2; TPL2, tumor progression locus 2; TRIB2, tribbles homologue 2.

**Table 3 cancers-15-00710-t003:** New treatments perspectives.

Preclinical Stage					
Drug Targets	Therapies	Thyroid Cancer Model	Experimentation Type	Effectiveness criteria	Ref
ARAF, BRAF, CRAF	TAK-632 vs. vemurafenib	3 ATC BRAF^V600E^ cell lines	- Quantification of MAPK pathway inhibition - Proliferation assay	TAK-632 > vemurafenib: On MAPK inhibition On GI50 and IC50	[56]
ARAF, BRAF, CRAF	LY3009120 vs. vemurafenib	- 3 PTC BRAF^V600E^ cell lines - Mouse xenograft model	- Viability assay - Apoptosis assay - Cytotoxic assay - In vivo tumor growth	1) LY3009120 overcame vemurafenib resistance due to BRAF-CRAF dimerization 2) LY3009120 > vemurafenib: On cell viability, toxicity and apoptosis On tumor growth inhibition in vivo	[58]
RAF + ERK1/2	Dabrafenib + SCH772984	- 5 BRAF^V600E^ cell lines (ATC + DTC) - Mouse xenograft model	- Quantification of MAPK pathway inhibition - Viability assay - Apoptosis assay - In vivo tumor growth	Dabrafenib + SCH772984 avoid MAPK reactivation observed with dabrafenib alone 1) Dabrafenib + SCH772984 > Dabrafenib or SCH772984 alone: On cell viability and apoptosis On tumor growth inhibition in vivo	[59]

Abbreviations: vs., versus.

**Table 4 cancers-15-00710-t004:** Preclinical studies of iodine recaptation in thyroid cancers models.

Drug Targets	Therapies	Thyroid Cancer Model	Experimentation Type	Effectiveness	Ref
BRAF	Vemurafenib (V) Dabrafenib (D)	3 PTC + 1 ATC BRAF^V600E^ cell lines	- NIS expression (RT-qPCR) - Iodide uptake assay - Gene expression scores related to TCGA derived gene signatures	Monotherapy (V) or (D): ↑ NIS mRNA ↑ Iodide uptake capacity ↑ Thyroid differentiation score (done in 1 PTC Cell line)	[66]
MEK	U0126	1 BRAF^V600E^ inducible rat thyroid derived cell line	- NIS expression (RT-qPCR)	↑ NIS mRNA	[67]
BRAF or MEK	Vemurafenib (V) Selumetinib (S) U0126 (U) CKI (C)	- 1 BRAF^V600E^ inducible rat thyroid derived cell line - Mouse model of BRAF^V600E^ PTC	Cell line: - NIS expression (WB) Mouse model experience: - NIS expression (RT-qPCR) - Iodide uptake assay - Tumoral response to RAI-therapy (tumor volume evaluated by US)	Cell line experience: ↑ NIS protein with (V), (S), (U), (C) Mouse model experience (C) vs. (S): ↑ NIS mRNA with (C) > (S) ↑ Iodide uptake capacity with (C) > (S) (knowing S > CTL) Tumoral response to RAI-therapy with (C) > (S) (knowing S > CTL)	[68]
BRAF + MEK	Dabrafenib (D) Trametinib (T)	- 1 PTC BRAF^V600E^ cell line - PTC-patient derived primary cell cultures	- NIS expression (RT-qPCR)	Cell line: No NIS re-expression with monotherapy (T) ↑ NIS mRNA with (D+T) PTC-Patient derived primary cell cultures: ↑ NIS mRNA with (T) Bi-therapy (D+T) even more efficient	[69]
BRAF + HDAC	Dabrafenib (D) Selumetinib (S) Panobinostat (P)	2 PTC BRAF^V600E^ cell lines	- NIS expression (RT-qPCR) - NIS localization (immunofluorescent microscopy) - Iodide uptake assay	Monotherapy (D) or (S): ↑ NIS mRNA ↑ NIS fluorescence to the cell membrane ↑ Iodide uptake capacity Bi-therapy (D+P) and (S+P) even more efficient on all experimentations	[70]
BRAF + EZH2	Dabrafenib (D) Selumetinib (S) Tazemetostat (T) (EZH2 inhibitor)	2 PTC BRAF^V600E^ cell lines	- NIS expression (RT-qPCR, WB) - NIS localization (immunofluorescent microscopy) - Iodide uptake assay	Monotherapy (D) or (S): ↑ NIS mRNA and protein ↑ NIS fluorescence to the cell membrane ↑ Iodide uptake capacity Bi-therapy (D+T) and (S+T) even more efficient on all experimentations	[71]
BRAF + HER	Dabrafenib (D) Selumetinib (S) Lapatinib (L)	2 PTC BRAF^V600E^ cell lines	- NIS expression (RT-qPCR, WB) - NIS localization (immunofluorescent microscopy) - Iodide uptake assay	(Monotherapy (D) or (S): ↑ NIS mRNA and protein ↑ NIS fluorescence to the cell membrane ↑ Iodide uptake capacity Bi-therapy (D+L) and (S+L) even more efficient on all experimentations	[72]
MEK + ACVR1B/TGFBR1	CKI (C) Vactosertib (V)	Mouse model of BRAF^V600E^ PTC	- NIS expression (RT-qPCR) - NIS localization (immunohistochemistry) - Iodide uptake assay	(C): ↑ NIS mRNA ↑ NIS fluorescence in tumors ↑ Iodide uptake capacity Bi-therapy (C+V) more efficient on Iodide uptake capacity but not on NIS mRNA and NIS fluorescence in tumors	[73]

**Table 5 cancers-15-00710-t005:** Clinical studies of redifferentiation strategies in radioactive iodine refractory thyroid cancers.

Drug Targets	Therapy (Duration of Treatment)	Thyroid Cancer Types	Oncogenic Driver	Study Design	N Total	Rate of RAI Uptake Restoration	RECIST Response (N Treated)	Ref
BRAF	Dabrafenib (6 weeks)	PTC	BRAF	- Prospective evaluation of RAI avidity restoration by diagnostic ^131^I-WBS - If avidity restored, treatment with fixed activity of 5.5 GBq	10	60%	At 3 months (n = 6): 2 PR, 4 SD	[75]
BRAF	Vemurafenib (4 weeks)	PTC	BRAF	- Prospective evaluation of RAI avidity restoration by diagnostic ^124^I PET-scan - If specific dosimetry criteria met, treatment with maximum tolerable activity (mean activity 9.4 GBq)	10	60%	At 6 months (n = 4): 2 PR, 2 SD	[76]
BRAF and/or MEK	- Dabrafenib +/− trametinib - Vemurafenib - Trametinib - Investigational MEKi (median 14 months, range 1–76.4)	77% PTC 15% PDTC 8% FTC	70% BRAF 23% RAS 7% WT	- Retrospective study including patients treated with MAPKi for RAIR-TC - Proof of RAI avidity restoration by ^131^I-WBS - Median administered activity: 7.5 GBq	13	62%	Median time of follow-up after RAI: 8,3 months (n = 8): 3 PR, 5 SD	[77]
BRAF and/or MEK	- Trametinib +/− dabrafenib - Vemurafenib + cobimetinib (4 weeks)	50% PTC 33% FTC 17% PDTC	50% BRAF 50% RAS	- Retrospective study including patients treated with MAPKi for RAIR-TC - Proof of RAI avidity restoration by ^124^I PET-scan - Mean administered activity: 7.9 GBq	6	67%	At 3 months (n = 4): 3 PR, 1 SD	[78]
MEK	Selumetinib (4 weeks)	65% PTC 35% PDTC	45% BRAF 25% RAS 15% RET/PTC 15% WT	- Prospective evaluation of RAI avidity restoration by ^124^I PET-scan - If specific dosimetry criteria met, treatment with maximum tolerable activity (NA mean activity)	20	60%	At 6 months (n = 8): 5 PR, 3 SD	[79]
BRAF + MEK	Dabrafenib + Trametinib (6 weeks)	PTC	BRAF	- Prospective evaluation of RAI avidity restoration by diagnostic ^131^I-WBS systematically followed by fixed ^131^I activity of 5.5GBq	21	Dc-WBS: 65% Pt-WBS: 95%	At 6 months (n = 21): 8 PR, 11 SD	[80]
BRAF + MEK	Trametinib +/− dabrafenib (3 weeks)	50% PTC 35% FTC 15% PDTC	70% WT 30% BRAF	- Prospective evaluation of RAI avidity restoration by diagnostic ^123^I-WBS - If avidity restored, treatment with mean ^131^I activity of 11 GBq	20	35%	Between 3–12 months (n = 7): 1 PR, 5 SD	[81]
BRAF + HER3	Vemurafenib + CDX-3379 (5 weeks)	50% PTC 50% PDTC	BRAF	- Prospective evaluation of RAI avidity restoration by ^124^I PET-scan - If specific dosimetry criteria met, treatment with maximum tolerable activity (mean activity 9.1 GBq)	6	83%	At 6 months (n = 4): 2 PR	[82]

Abbreviations: Dc-WBS, diagnostic whole-body scan; N, number of patients; Pt-WBS, post-therapeutic whole-body scan.

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
