# Peer review of "MAPK Pathway Inhibitors in Thyroid Cancer: Preclinical and Clinical Data"

_cancers, 2023, doi:10.3390/cancers15030710_

Round 1

Reviewer 1 Report

This is a very comprehensive review paper that could benefit the researchers in both MAPK-related TKI development and TC clinical study. The paper also discussed the redifferentiation strategy in TC therapeutic area which also gain lots of incremental information for the paper.

My only comment is, although there are so many TKIs can be used for TC treatment, thyroidectomy is still the first-line treatment of TC. I would like to know your thoughts on this. Thanks

Author Response

Dear reviewer,

Thank you for your review of our paper, and for the comments.

Thyroidectomy is indeed the first line treatment for thyroid cancer. Even for refractory thyroid cancers, thyroidectomy remains necessary to propose a redifferentiating treatment in order to retreat with radio-iodine. In addition, cervical surgery is the firstline treatment for locoregional recurrence.

Thanks again.

Best regards, Louis Schubert

Reviewer 2 Report

The manuscript is a comprehensive review of fundamental scientific discoveries and clinical trials of BRAF, MEK, and ERK inhibition and a discussion of resistance to MAPK pathway inhibitors and redifferentiation therapy. The review is well-written and enjoyable to read.

I have only minor comments.

Typos:

Line 137: capitalize RAS. Line 159, The Sentence on line 330 needs rewording.

Line 467. Response and not “refixation”?

The limitations of the redifferentiation clinical trials highlighted in Table 5 are short follow-up (<= 6 months for many studies) and lack of randomization. It is unclear if RECIST response is durable or even significant should it be compared to observation or TKI treatment alone. These limitations are addressed by the authors and further supported by the lack of effectiveness in the ASTRA trial.

The authors chose not to cover newly developed RET and TRK inhibitors, likely the most exciting development in treating advanced thyroid cancer in the last few years. These drugs inhibit MAPK by blocking oncogenic fusions upstream of the BRAF. I trust that these drugs were considered out of scope for this review. 

Author Response

Dear reviewer,

Thank you for your careful review of our paper, and for the comments, corrections and suggestions that ensued.

In fact, RET and NTRK inhibitors are very promising. They were considered out of scope for this review because we focused on the principal proteins of the MAPK pathway (RAS, RAF, MEK, ERK). Nevertheless, it would be interressant and worthwhile to do a specific review on this topic.

Thanks again

Best regards, Louis Schubert